# Addressing Sexual Violence Against Persons with Disabilities in Belgium

**DOI:** 10.3390/healthcare13233125

**Published:** 2025-12-01

**Authors:** Elizaveta Fomenko, Lotte De Schrijver, Anne Nobels, Ines Keygnaert

**Affiliations:** 1International Centre for Reproductive Health (ICRH), Violence Research, Capacity Building, and Care (VIORESC), Department of Public Health and Primary Care, Ghent University, 9000 Ghent, Belgium; lotte.deschrijver@vvkp.be; 2Vlaamse Vereniging voor Klinisch Psychologen, 1000 Brussels, Belgium; 3Department of Psychiatry, Ghent University Hospital, 9000 Ghent, Belgium; anne.nobels@ugent.be; 4Department of Geriatrics, Ghent University Hospital, 9000 Ghent, Belgium; 5Women’s Clinic, Ghent University Hospital, 9000 Ghent, Belgium

**Keywords:** sexual violence, persons with disabilities, vulnerability, mental health, ableism, socioeconomic inequality, victimization, Belgium, prevention, inclusion

## Abstract

**Highlights:**

**What are the main findings?**

**What are the implications of the main findings?**

**Abstract:**

**Background/Objectives**: This study examined the specific vulnerabilities and experiences of persons with disabilities (PwDs) regarding sexual violence (SV) in Belgium. **Methods**: Data were drawn from the nationally representative UN-MENAMAIS survey (n = 4944), which included adults aged 16–99 years. SV was assessed using behaviourally specific questions based on validated international instruments. Logistic regression analyses examined associations between SV, disability status, sociodemographic, and mental health indicators. **Results**: PwDs reported a significantly higher lifetime prevalence of hands-on SV (37.8%) compared with those without disabilities (29.4%; *p* < 0.001, V = 0.065). PwDs also reported lower quality of life (*p* < 0.001, V = 0.273), and higher rates of depression (*p* < 0.001, V = 0.214), anxiety (*p* < 0.001, V = 0.145), PTSD (*p* < 0.001, V = 0.101), sedative use (*p* < 0.001, V = 0.237), and suicide attempts (*p* < 0.001, V = 0.124), though they reported less hazardous alcohol use (*p* < 0.001, V = 0.103) and cannabis use (*p* < 0.001, V = 0.080). Regression analyses showed that individuals assigned female at birth (*p* < 0.001) and those identifying as LGB+ (*p* < 0.001) were at a higher risk of both hands-off and hands-on SV. Among mental health correlates, depression, anxiety, PTSD, substance use, self-harm, and suicide attempts were significantly (all *p* < 0.001) associated with increased odds of SV. **Conclusions**: The findings highlight the need for targeted, inclusive prevention and support strategies addressing structural inequalities, ableism, and barriers to care to effectively prevent SV and revictimization among PwDs.

## 1. Introduction

Sexual violence (SV) is defined as “every sexual act directed against a person’s will, by any person regardless of their relationship to the victim, in any setting” [1]. It consists of sexual harassment without physical contact—further called hands-off SV—and sexual abuse with physical contact but without penetration and (attempted) rape with penetration, further referred to as hands-on SV [2,3]. SV affects people worldwide, in every culture, and in every social layer of society and is considered a major public health issue [4]. Also in Belgium, SV was identified as an important threat to public health. According to the UN-MENAMAIS study (UNderstanding the MEchanisms, NAture, MAgnitude and Impact of Sexual violence in Belgium) [3], which surveyed a representative sample of Belgian citizens between 16 and 69 years old, it was found that 64% of the population has experienced some form of SV at least once in their lives. More precisely, 44% reported experiencing hands-on or hands-off SV within the past year [3,5]. Belgium ratified the Istanbul Convention in 2016 [6], committing to combat violence against women and domestic violence. In 2020, the Flemish government prioritized tackling SV—with attention to the most vulnerable groups, including persons with disabilities (PwDs), through a national action plan [7]. A 2018 qualitative study described the vulnerability of women with disabilities to SV in Flanders [8]. While the results should be approached with caution due to the exploratory nature of the study, they indicate a very high risk of SV for women with disabilities in Flanders. The study found that 93% of participants had experienced some form of SV. It was also noted that these victims often experienced repeated incidents, with the assailant being someone they knew, such as a partner, friend, or family member [8].

SV emerges and continues to exist due to factors and dynamics occurring at individual, interpersonal, community and societal level [9,10]. At individual-level risk and protective factors for SV, include younger age, being female and/or identifying as a woman, lower educational and/or socio-economic status, physical or mental health issues, dependence on others for care, engaging in risky behaviours (such as alcohol abuse, drug use, and unsafe sexual activity), and previous (in)direct exposure to violence [3,5,10,11,12,13,14,15,16,17,18,19,20]. On interpersonal, community, and societal levels, ruling gender norms, gender inequality, ideologies about male sexual entitlement, rape myth acceptance, insufficient legal frameworks targeted at sanctioning sexual perpetration and protecting victims of gender-based violence, etc., are identified as drivers for creating contexts that promote and sustain SV in varying degrees [10,21,22]. Although everyone can be impacted by these factors and is at risk of sexual victimization and perpetration, specific subgroups are more vulnerable to its exposure. The previously mentioned UN-MENAMAIS study [3] illustrated that not only applicants for international protection and lesbian, gay, bisexual, and other non-heterosexual (LGB+) persons [23] are particular at high risk of sexual victimization, but also that persons who identified as having characteristics that would differentiate them from the majority of the inhabitants of Belgium report more SV exposure than those who do not identify as such [24,25]. Hence, it was concluded that populations exposed to societal othering, which are often considered minority groups, experience a heightened victimization risk [25]. As a result of exposure to stigma, prejudice, and discrimination—they are susceptible to differential treatment in the societies in which they live [26]. Hence, they often hold a more vulnerable social position that increases the likelihood that they will present any of the above-mentioned general risk factors (cf. supra)—aside from risk factors specific to the othered group they belong to [25]. ‘Othering’ refers to processes that serve to mark and name those individuals considered as different from oneself and which secure and define a person’s or group’s identity through the stigmatization and distancing of others through “us-them” separations [27]. Moreover, not only do othered people experience more vulnerabilities for sexual victimization, they also often report help-seeking barriers that prevent them from finding adequate care to cope with its consequences and to prevent revictimization in the long run, and when they do access care, they often receive suboptimal or inadequate support [25,28,29]. Furthermore, when people have a combination of multiple othered identities, the impact and probability of sexual victimization increases [24,25]. It is therefore crucial to apply the framework of intersectionality [30] when studying SV, as multiple minority identities may yield different social experiences and subsequently also risk factors.

With the intention to combat SV in the Belgian society, the Agency for Home Affairs of the Flemish Government requested to explore the specific situation of persons with disabilities (PwDs) more in-depth, as they are identified as a potentially vulnerable group at increased risk of sexual victimization. In line with the UN Convention on the Rights of Persons with Disabilities [31], we define PwDs as persons who have long-term physical, mental, intellectual, or sensory impairments which, in interaction with various barriers, may hinder their full and effective participation in society on an equal basis with others. PwDs are, for example, recognized as a group that often reports a lower socio-economic status and economic poverty, ableism (i.e., stigma, prejudice, and discrimination related to disabilities), poor health and well-being statuses [32,33], and they might often depend on others for care and assistance with daily activities; factors which are identified as increasing the risk of sexual victimization. In this study, we refer to such conditions as structural vulnerabilities, that is, socially and institutionally embedded disadvantages that systematically constrain individuals’ access to resources, protection, and opportunities. These vulnerabilities are produced and reinforced by societal structures rather than by individual characteristics, thereby increasing exposure to SV and reducing access to support. Indeed, a recent meta-analysis by Mailhot Amborski et al. [34] showed that PwDs are at higher risk of sexual victimization than people without disabilities (OR = 1.49; 95% CI [1.27, 1.76]). Upon exploration of potential moderators, they found that both minor (age < 21) and adult (age = 21+) PwDs show this trend, but adult PwDs were found to be more at risk of sexual victimization than minor PwDs [34]. This finding is in contrast with what is generally found in SV studies, namely that the risk of sexual victimization increases with younger age (see e.g., [5]). In addition, the type of disability also emerged as a moderator in this meta-analysis [34]. Persons with intellectual deficits, physical disabilities, sensory disabilities (=highest risk), as well as those with multiple co-occurring disabilities spanning more than one type (e.g., a combination of cognitive and physical impairments), were all significantly more at risk than the general population [34]. In this meta-analysis, psychological or emotional disability was not identified as a significant moderator. The available evidence regarding SV against PwDs also highlights important help-seeking barriers experienced by the victims. Several studies have shown that SV within this population is severely under-reported and that when victims do disclose and report SV, their experiences are often ignored, dismissed, downplayed, and concealed [8,35,36,37]. Disclosure and help-seeking require recognizing sexual transgressive behaviour and knowing where to seek assistance [38]. Identification can be challenging for PwDs as they often depend on others and may face power imbalances in their relationships, making it harder to recognize abuse [37]. Moreover, especially for people with cognitive impairments or intellectual deficits, recognizing transgressive behaviour as violence and reporting SV may be difficult [8,35,37,39]. As a result, SV against PwDs often remains under the radar, and victims do not receive the needed care upon victimization.

With this study, we aim to (1) estimate the prevalence of SV in PwDs in Belgium and (2) compare SV rates with those reported by participants without disabilities. We hypothesize that PwDs will be more likely to be exposed to SV during their lifetime. We will also explore (3) vulnerabilities associated with SV. More explicitly, we will study whether the correlates of sexual victimization observed in the general population (i.e., sex at birth, age, socio-economic status, mental health status, and coping strategies) are associated with and potentially moderated by having disabilities.

## 2. Materials and Methods

### 2.1. Sampling Procedure and Participants

This study formed part of a broader mixed-methods research project called ‘UNderstanding the MEchanisms, NAture, MAgnitude and Impact of Sexual violence in Belgium’ (UN-MENAMAIS) [3]. The main aim of this project was to investigate sexual victimization and perpetration among a randomly selected sample of Belgian residents, regardless of their gender or sexual orientation, using a pre-validated self-report.

#### 2.1.1. Belgian Citizens Aged 16 up to 69 Years Old

In a cross-sectional quantitative study, an online survey was conducted to collect data from a nationally representative sample of Belgian citizens aged 16 to 69 years. The survey was conducted in two waves, from October 2019 to January 2021, using the Belgian National Register (BNR) as the sampling frame. To ensure equal representation of male and female participants, a random disproportionate stratified sampling method was employed, with participants divided into three age groups (i.e., 16–24 years old, 25–49 years old, and 50–69 years old). The initial overrepresentation in the first wave was adjusted in the second wave using survey weights to obtain estimates representative of the Belgian population (see [5] for more details). A total of 41,520 Belgian residents between 16 and 69 years old were contacted by the BNR through the post, and participants could access the survey through a link or a Quick Response (QR) code with informed consent. The online survey was started by 6504 respondents.

#### 2.1.2. Belgian Citizens Aged 70 Years and Older

From 8 July 2019 to 12 March 2020, a cluster random probability sampling with a random walk finding approach [11] was used to select a representative sample of older adults living in Belgium (with clusters referring to randomly selected geographical areas across the country) to participate in structured face-to-face interviews. To participate in the study, participants had to be at least 70 years old, reside in Belgium, and be able to complete the interview in Dutch, French, or English. Cognitive ability was assessed through consistency in answering questions and attention during the interview. Interviews were conducted by trained interviewers at the participant’s residence (i.e., nursing home, assisted living facility, or community). A total of 513 interviews were performed (i.e., 34% participation rate) [11].

### 2.2. Measures

#### 2.2.1. Assessment of Sexual Violence

In this study, SV was defined according to the WHO definition (cf. supra) [1]. As is recommended internationally, behaviourally specific questions [40,41,42] were used to provide reliable estimates of both female and male sexual victimization [42], for participants of different sexual orientations or gender identity, or different cultures. The details of the validation procedure are described elsewhere (see [42]).

The questionnaire was designed to maximize SV (victimization and perpetration) disclosure by starting with less sensitive topics, building up towards the questions regarding SV. Both lifetime and past 12 months SV experiences were assessed. The behaviourally specific questions were derived from the revised Sexual Experience Survey (SES-R) [41,43], the Sexual Aggression and Victimization Scale (SAV-S) [44], and the Senperforto questionnaire [45]. All questions on SV were adapted to the Belgian social and legal context. The process of developing this survey has been described elsewhere (see [3,42]).

#### 2.2.2. Assessment of Socio-Economic Status

To explore the socio-economic status, participants were asked about their highest level of education (i.e., I didn’t go to school; primary education; secondary education; technical and vocational education (apprenticeship); religious school (e.g., madrassa); or higher education (college/university education leading to a bachelor’s or master’s degree)), their current occupational situation through the question “What best describes your current situation?” (i.e., student; housewife/man; voluntary work; on the job market/looking for a job; employed/self-employed; contributing family member; not able to work because of ill health; financial self-sufficiency or any other type of alternative choice of living; retired; or other), and their occupational situation before retirement through the question “What describes your situation before you retired?” (i.e., housewife/man; voluntary work; on the job market/looking for a job; employed/self-employed; contributing family member; not able to work because of ill health; financial self-sufficiency or any other type of alternative choice of living; or other). We created a new variable ‘able to work’ by combining the current occupational situation and the occupational situation before retirement. If participants indicated ‘not able to work because of ill health’ in at least one of these two variables, they were coded as ‘not able to work (1)’. Others were coded as ‘able to work (0)’. Finally, the financial situation was assessed by asking “Considering your monthly income as a household, would you say that your household is able to make ends meet…” and proposing the answer options ‘with great difficulty’, ‘with some difficulty, ‘fairly easy’, and ‘easily’. The first two answer options were regrouped into ‘financial situation perceived as difficult’ and the latter two into ‘financial situation perceived as easy’.

#### 2.2.3. Assessment of Disability and/or Chronic Illness

To identify the PwDs subgroup within the study sample, two survey items were used. Participants were asked “Do you suffer from a chronic illness that limits you in your everyday activities?” and “Do you suffer from a disability that limits you in your everyday activities?”. Everyday activities were defined as “for example, working, shopping, going to school, managing your life, keeping in contact with other people”, which was added beneath both questions. Participants who indicated ‘Yes’ for one or both questions were coded as having a ‘disability and/or chronic illness (1),’ and participants who indicated ‘No’ on both questions were coded as ‘no disability and/or chronic illness (0)’. Many participants in the study had difficulty differentiating between a disability and a chronic illness, resulting in the terms being used interchangeably. Participants were asked to provide an explanation of their disability and/or chronic illness in an open-ended format. Although disability and chronic illness are conceptually distinct, in practice, the responses revealed substantial overlap between the two. Many participants interpreted the questions subjectively and used the terms interchangeably, providing open-ended descriptions that could not always be clearly classified as either disability or chronic illness. No strict definition was provided in the survey, and respondents answered to the best of their understanding. As a result, conditions identified as a disability by one participant could be framed as a chronic illness by another, and vice versa. Because these categories were intertwined, both were combined into a single variable to capture the broader group of individuals reporting functional limitations in everyday life. The responses were reviewed by two researchers and co-authors of the study, as well as a general practitioner. All participants were included in the new variable, as the reported disabilities and/or chronic illnesses could potentially be classified as disabilities at some point in their progression. However, it is uncertain whether the participants reporting these conditions are currently experiencing hindrances in their full and equal participation in society. Therefore, the assessment of their status as PwDs is subjective.

Finally, we created an additional variable through the combination of ‘PwDs’ and ‘able to work’. Participants who did not indicate any disability or chronic illness and were coded as ‘no disability and/or chronic illness (0)’ remained as such. Participants who were coded as ‘disability and/or chronic illness’ were further divided into ‘disability and/or chronic illness, but able to work (1)’ (if they were coded as ‘able to work’) and ‘disability and/or chronic illness, but not able to work (2)’ (if they were coded as ‘not able to work’). In the Belgian context, the status of being unable to work is often medically certified (work incapacity) and typically follows a medical assessment by a physician. However, in this survey, we could not verify whether each respondent’s reported inability to work was officially certified or directly linked to their disability or illness. Therefore, this variable should be interpreted as an approximate proxy for functional impairment that may, but does not necessarily, correspond to medically recognized work incapacity. Given that our PwDs group is highly heterogeneous and based on self-reported disability, this operationalization allowed us to distinguish a subgroup likely to experience more objectively verifiable functional limitations from those whose disability experience may be more subjectively defined, and it may also tentatively reflect differences in severity, as individuals who are unable to work often contend with more substantial or impactful impairments or chronic conditions.

#### 2.2.4. Assessment of Coping Strategies and Mental Health Status

Specific mental health aspects were measured in all participants by validated scales. Depression was assessed using the 9-item Patient Health Questionnaire (PHQ-9) [46]. Responses were made on a 4-point Likert scale ranging from ‘not at all (0)’ to ‘nearly every day (3)’. All items were summed in a final score ranging from 0 to 27, Cronbach’s Alpha = 0.867. Anxiety was measured by the General Anxiety Disorder (GAD)-7 [47]. The scale had seven items, and responses were made on a 4-point Likert scale ranging from ‘not at all (0)’ to ‘nearly every day (3)’, Cronbach’s Alpha = 0.888. All items were summed in a final score ranging from 0 to 21 to yield a total anxiety score. Both scales assessed symptoms in the two weeks prior to filling in the survey, and both used a cut-off score of five as a positive screening for depression and/or anxiety [46,47].

Posttraumatic Stress Disorder (PTSD) was measured using the PC-PTSD-5 (Cronbach’s Alpha = 0.833), which asked about symptoms in the month before completing the survey [48]. On this scale, with five items with a response format of ‘yes (1)/no (0)’ answers, a score of three or higher of a maximum of five was regarded as an indication of PTSD [48].

Quality of life was assessed via a 5-point Likert scale ranging from ‘very poor (1)’ to ‘very good (5)’ with the question “How would you rate your quality of life?”.

Resilience was assessed using the 6-item Brief Resilience Scale (BRS) (Cronbach’s Alpha = 0.938). Responses were made on a five-point scale ranging from 1 (=strongly disagree) to 5 (=strongly agree). All six items were averaged into a final score ranging from 1 to 5 [49].

To assess maladaptive coping strategies generally associated with SV, we investigated alcohol and drug use, self-harming behaviour, and suicide attempts. Hazardous alcohol use was screened for using the AUDIT-C [50] (Cronbach’s Alpha = 0.690). The AUDIT-C consists of three questions: “How often do you have a drink containing alcohol?” ranging from ‘Never (0)’ to ‘4 or more times a week (4)’ (the screening ends with a score of 0 for respondents that indicated ‘Never’ in this first item), “How many standard drinks containing alcohol do you have on a typical day” ranging from ‘1 or 2 (0)’ to ’10 or more (4)’ and “How often do you have six or more drinks on one occasion?” ranging from ‘Never (0)’ to ‘Daily or almost daily (4)’. In accordance with the guidelines of ‘Flemish centre of expertise on alcohol and other drugs (VAD)’, a cut-off score of four for females and five for males was used on this 3-item scale, with a total score between zero and 12 [50]. In addition to the validated scales, participants were asked, using yes-no questions, about sedative use, cannabis use, illegal drug use, self-harm, and suicide attempts, both during their lifetime and in the past 12 months. Responses were categorized as ‘No (0)’, ‘Yes, during the lifetime, but not in the past 12 months (1) and ‘Yes, during the past 12 months (2)’.

### 2.3. Ethical Considerations

This study was designed and performed in line with the principles of the Declaration of Helsinki and was approved by the Commission for Medical Ethics of Ghent University Hospital/Ghent University (B670201837542). Only participants aged 16 and older were included in this study due to ethical and practical regulations regarding the legal age of sexual consent in Belgium (16 years old), as confirmed by the ethical committee.

All participants provided informed consent before starting the survey. For the online survey, participants were required to check multiple boxes in Qualtrics, explicitly agreeing to multiple statements. If any box was left unchecked, participants were unable to proceed with the survey. While this process was not directly witnessed for online participants, the responses, including the confirmation of consent, were systematically documented in the dataset. For face-to-face interviews with elderly participants, informed consent was obtained and directly witnessed by the interviewer. As no participants under the legal age of consent (16 years) were included in the study, additional parental or guardian consent was not required.

### 2.4. Analysis

All analyses were run in R4.1.1. Descriptive statistics (medians, interquartile ranges, counts, and percentages) were computed for all variables across all tables. Significant differences in the distribution of nominal or categorical variables between (1) No PwDs and PwDs, and between (2) PwDs who were able to work and PwDs who were not able to work, were computed using (post hoc) chi-square tests. If the assumptions were not met, Fisher’s Exact test was used. No independent samples t-tests were used as none of the continuous variables were normally distributed. Two binary logistic regressions (one for hands-off SV and one for hands-on SV) were used to analyze the association between socio-demographic variables, mental health and well-being, and the prevalence of lifetime hands-off and hands-on SV. To avoid multicollinearity, the correlations between all variables were checked. Variance Inflation Factors (VIF) were calculated and indicated no problematic multicollinearity. Each logistic regression was conducted in two steps. In a first step, all socio-demographic and mental health variables were entered as main effects, after which interaction terms (cross-products) between each predictor and the three-level disability variable (‘no disability’, ‘yes, but able to work’, ‘yes, but not able to work’) were added to this full model. As none of the interaction terms reached statistical significance (*p* < 0.05), we proceeded to a second step in which only the main effects were retained and combined in one final full model. The independent variables included in the models were selected based on previous literature and in line with the structural vulnerabilities discussed in the Introduction (e.g., socio-economic position, mental health, coping behaviour) as well as our research question. Importantly, the logistic regressions were not fitted as predictive models, nor was prediction the aim of these analyses. Instead, they were used to explore associations and overall trends between relevant vulnerability factors and SV outcomes, providing insight into how these factors relate to each other within our sample rather than to build or validate predictive tools.

## 3. Results

### 3.1. Sample

We gathered data from 7017 respondents. Respondents were excluded for the following reasons: lack of informed consent (n = 706), incomplete survey responses (n = 909), not meeting the age criterion (i.e., younger than 16 years; n = 6), duplicate survey submissions (n = 37), and concerns about response quality (n = 1). Additionally, respondents with missing values on key variables for this study (e.g., items related to disability assessment) were excluded (n = 414).

The final analysis included 4944 observations. This sample comprised 4461 individuals aged 16 to 69 from the general population and 483 older adults aged 70 and above. Among the participants, 2427 were assigned male at birth, and 2517 were assigned female at birth. The average age of the sample was 42.83 years (SD = 20.15). A majority of participants (89%) were born in Belgium. The survey was completed in Dutch 3048 times, in French 1732 times, in English 150 times, in Arabic nine times, and in Farsi five times.

Table 1 summarizes sociodemographic characteristics of the unweighted sample. PwDs differ significantly from non-disabled participants in terms of sociodemographics. PwDs had more female participants, older age, lower education, less employment, more financial difficulty, and higher self-identification as LGB+. Differences between PwDs unable to work and those able to work were smaller. Incapacitated PwDs were younger and had more financial difficulties.

The study sample overrepresents higher educated individuals compared to the general Belgian population. Almost half of all respondents (i.e., 48.4%) completed a level of higher education, while—on the population level—37.6% of Belgian residents between 15 and 64 years completed a higher educational level [51]. Table 2 presents the comparison of men and women across age groups in the entire population (ages 16–99) using public data and our sample.

### 3.2. Mental Health, Quality of Life and Well-Being

Table 3 compares mental health, quality of life, and well-being in PwDs with individuals without disabilities or chronic illnesses. It also compares these variables between PwDs who are unable to work and those who can.

PwDs experienced worse mental health, quality of life, and well-being compared to non-disabled individuals, regardless of their history of SV. PwDs, both with and without SV, reported lower quality of life, more symptoms of depression, anxiety, PTSD, sedative use, and suicide attempts compared to those without disabilities. However, there was significantly less hazardous alcohol and cannabis use among PwDs. No significant differences were found in resilience, illegal drug use, and self-harm between PwDs and non-PwDs individuals.

Among PwDs, those unable to work reported lower quality of life, more symptoms of depression, anxiety, and PTSD, and higher sedative use.

### 3.3. Prevalence of Sexual Violence

Table 4 shows the prevalence of hands-off and hands-on SV in the total sample and among those with disabilities. PwDs experienced higher rates of hands-on SV compared to those without disabilities or chronic illnesses, but the rates of hands-off SV were similar in both groups.

Although there was only one significant difference (attempt of vaginal or anal penetration) after applying a strict Bonferroni correction, we still see a clear (marginally) significant difference between PwDs who can work and those who cannot. PwDs who are incapacitated show a greater proportion of exposure to multiple forms of hands-off and hands-on SV compared to those who can work.

No moderating effect of disability was found in the relationship between the socio-demographic or mental health variables and sexual violence. Table 5 shows the findings of the two logistic regressions. Overall, the predictors included in the models explained 22.8% of the variance in hands-off SV and 14.2% of the variance in hands-on SV.

Socio-demographic variables improved both models significantly, except for educational level in both hands-off and hands-on SV, and reporting a disability or chronic illness in hands-off SV. However, significant differences were found in participants’ sex assigned at birth, age, and sexual orientation. Individuals assigned female at birth and/or self-identified as being LGB+ had a higher risk of both hands-off and hands-on SV. Participants over 50 had a lower risk of hands-off SV, and those between 25 and 49 had a lower risk of hands-on SV compared to those aged 16–24.

Strong correlations were found for mental health and well-being. All mental health factors improved both models, except for quality of life, resilience, and illegal drug use. People who reported higher anxiety and/or PTSD symptoms, problematic alcohol and sedative use, cannabis use, suicide attempts, and self-harm were more at risk of hands-off and hands-on SV.

## 4. Discussion

This is the first study in Belgium to estimate the prevalence of SV against PwDs using nationally representative data. International research reports prevalence estimates that vary widely depending on the type of disability studied and the definition of SV applied, making direct comparisons challenging [34]. However, the findings confirm that PwDs—consistent with previous studies (see, e.g., [8])—are more vulnerable to SV compared to people without disabilities, particularly when it comes to hands-on SV. However, in contrast to international meta-analytic findings that indicate approximately a twofold higher risk of SV among PwDs compared to non-disabled individuals [34], the difference observed in our study appears smaller. This may be explained by the heterogeneity of our PwDs group, which includes individuals with both disabilities and chronic illnesses, as well as by the exclusion of persons with severe intellectual or mental disabilities for whom participation was not feasible. Additionally, PwDs who are unable to work tend to experience higher levels of both hands-off and hands-on SV compared to those who can work, although the differences may be small. This trend suggests that individuals who rely on others for care, housing, safety, etc., and those who have financial constraints are more vulnerable to sexual victimization [8,19,25,37,52,53,54,55]. Moreover, the apparent sample differences in terms of socio-demographic characteristics, mental health and coping outcomes, and the applied logistic regression analyses reveal that—as observed in other vulnerable groups that are often exposed to social othering [23,24]—the observed higher prevalence of SV in PwDs can be explained by the increased likelihood that they hold a more vulnerable social position rather than this increased risk being associated with their specific minority characteristic—in this case disability—per se. In this line, we found that the general risk factors for SV such as having a female sex assigned at birth, having a younger age, identifying as LGB+, worrying about one’s financial situation, reporting poor mental health, hazardous alcohol use, sedative and cannabis use, self-harming behaviour and suicide attempts were key to significantly optimize the prediction of SV in PwDs [5,13,23,56,57,58,59]. Yet, many of these factors were also more common among PwDs in our sample. Moreover, as expected based on the literature and identified as risk factors for increased SV exposure [12,20,60,61,62,63], PwDs, in general, reported poorer mental health, quality of life, and well-being than study participants without disabilities or chronic illnesses. However, PwDs reported less hazardous alcohol and cannabis use and no differences between PwDs and participants who do not report disabilities or chronic illness were found for illegal drug use and self-harming behaviour. This is an interesting finding as these variables were shown to significantly increase the predictive value of our model. Yet, in our logistic regression model, having a disability did not show to have a significant interaction effect on the relationship between mental health and victimization. Furthermore, in contrast to earlier studies [34], we could not confirm that adult PwDs were more at risk than minor PwDs. However, this can potentially be explained by our grouping of both respondents reporting disabilities and chronic illnesses into one PwDs variable. Chronic diseases have been identified both among the consequences as well as among the risk factors of sexual victimization and often emerge in later life [29]. It is therefore likely that in our sample, with age the likelihood of a respondent having experienced SV and being at risk of revictimization increased.

Our findings suggest that the higher prevalence of SV in PwDs is not so much related to the mere fact of having a disability, but rather to underlying factors that increase the likelihood of SV. In line with our results, these underlying factors appear to reflect the structural and social processes of othering and ableism, through which PwDs are marginalized and systematically deprived of equal access to economic stability, autonomy, social support and adequate care upon sexual victimization. Specifically, poorer mental health, financial hardship, social isolation, and dependency on others for daily care, for housing and for accessing external professional care may heighten vulnerability to coercion and limit their ability to avoid or report SV or receive adequate care upon victimization. These interrelated disadvantages reinforce each other and illustrate how social exclusion, rather than disability itself, creates the conditions in which sexual violence can occur and persist. Future research should uncover the causes and interplay of these risk factors to identify key elements for effective SV prevention.

### 4.1. Limitations and Suggestions for Future Research

Limitations exist in our study that should be addressed. Firstly, our sample may not accurately represent the general Belgian population (cf. sample 16–69 years old) in terms of educational level and language distribution, despite using random recruitment methods. This could introduce bias. Additionally, the overrepresentation of Flemish speaking participants suggests a potential regional imbalance among our participants, possibly leading to cultural differences across Belgian regions that may have influenced our findings. Moreover, differences in data collection strategies across age groups, specifically the use of online surveys for participants aged 16–69 and structured face-to-face interviews for those aged 70 and above, may have influenced reporting patterns and prevalence rates of SV. Such methodological variation could affect comparability across age ranges and should be interpreted with caution. Secondly, supported by the available literature (see, e.g., [34]), we recognize that type and degree of disability or chronic illness are potentially significant moderators. Because of our data collection design, we could not control for varying degrees of disability or types of disabilities, which may have affected the identification of vulnerabilities for sexual victimization in PwDs. Furthermore, disability status in our study relied on participants’ self-assessment (“subjective” reporting) rather than on medical or administrative verification. This approach may introduce reporting bias and should therefore be considered a limitation. Future studies should use better-balanced samples and consider factors like type and degree of disability, residence in a facility, and professional care received. Population studies on SV need large samples to compare different types of disabilities and identify specific risk factors related to long-term impairments. Finally, it is important to acknowledge that the study did not explicitly assess the capacity for informed consent among participants with intellectual or severe mental disabilities in the online survey administered to respondents aged 16–69 years. Although all participants were informed about the study aims and consent procedures before participation, it is possible that some individuals with cognitive impairments in this group may not have been fully able to provide informed consent in practice. In contrast, for the structured face-to-face interviews with participants aged 70 years and older, interviewers were trained to assess understanding and consent capacity prior to data collection. While we consider the overall risk of compromised consent to be limited given these procedures, this nonetheless represents a potential ethical and methodological limitation that should be considered when interpreting the findings. Future research should also consider the intersectionality of disability with other characteristics such as gender, sexual orientation, and ethnicity to explore increased victimization rates among PwDs who belong to multiple othered groups. It is important to note that the exclusion of individuals with severe mental disabilities in this study underestimates the reality.

### 4.2. Care, Prevention, and Policy Implications

To effectively break the circle of SV and revictimization [25], tailored prevention strategies across multiple levels are essential. Policymakers should prioritize research to understand the prevalence and determinants of SV in PwDs, focusing on factors such as dependency, housing and care facilities for PwDs, financial insecurity, and poor mental health. Prevention programmes must be inclusive and accessible, using, e.g., visual aids, plain language, and interactive workshops to raise awareness about sexual consent and SV. Moreover, healthcare and social care providers should receive mandatory diversity sensitive training to recognize vulnerabilities and screen for and act adequately on SV risks, particularly in individuals presenting with poor mental health, substance use, or self-harm.

Enhanced victim support services are crucial to mitigating the impact of SV, including ensuring that 24/7 helplines, psychological counselling, and legal assistance are better tailored to PwDs needs as well. Interventions addressing risk factors like work incapacitation and unstable housing are necessary, alongside targeted social programmes offering financial and psychosocial support. Additionally, inclusive legal frameworks should simplify reporting processes and provide free legal aid to PwDs, ensuring their protection and access to justice.

Broader societal change is equally critical. National campaigns to combat ableism and promote inclusivity can address the stigma, prejudice, and discrimination that exacerbate PwDs’ vulnerability to SV. Collaborative efforts among governmental bodies, disability organizations, and educational institutions are essential for implementation. Finally, integrating these measures into existing health, social, and legal systems can ensure a coordinated and sustainable approach to preventing SV and supporting PwDs. By addressing structural vulnerabilities and fostering inclusivity, these measures offer a pathway to breaking the cycle of SV and ensuring a safer, more equitable society for PwDs.

## 5. Conclusions

This study provides the first nationally representative evidence on SV against PwDs in Belgium. PwDs face a significantly higher lifetime prevalence of hands-on SV than individuals without disabilities. These differences are largely explained by intersecting structural vulnerabilities, including financial hardship, dependence on care, and poorer mental health, rather than disability itself.

The findings underscore that SV against PwDs constitutes a critical and preventable health disparity rooted in social determinants such as ableism, economic exclusion, and inequitable access to protection and care. Addressing these determinants requires multisectoral action: inclusive prevention education, professional training for healthcare and social service providers, and stronger legal and social support frameworks.

Future research should further disentangle the role of disability type and severity, and examine how intersecting identities (e.g., gender, sexual orientation, ethnicity) compound risk. By integrating disability-inclusive approaches into public health and social policy, Belgium, and other countries can move toward reducing health inequalities and ensuring equitable safety and well-being for all.

## Figures and Tables

**Table 1 healthcare-13-03125-t001:** Sample composition (n = 4944). Socio-demographic information was presented for persons with disabilities (PwDs) and persons without disabilities within the total study sample.

Variable	Within Total Sample (n = 4944)			Within Group Disability (n = 716)		
	No Disability (n = 4228; 85.5%) n (%)	Disability (n = 716; 14.5%) n (%)	χ^2^; df; *p*-Value; V	Able to Work (n = 587; 82.0%) n (%)	Unable to Work Due to Disability (n = 129; 18.0%) n (%)	χ^2^; df; *p*-Value; V
**Sex assigned at birth**			12.36; 1; <0.001; 0.050			0.24; 1; 0.625; 0.018
Male	2119 (50.1)	308 (43.0)		255 (43.4)	53 (41.1)	
Female	2109 (49.9)	408 (57.0)		332 (56.6)	76 (58.9)	
**Age [mean (SD)]**	40.71 (19.21)	55.33 (21.05)	462.79; 3; <0.001; 0.306	56.11 (33.66)	51.77 (11.44)	103.35; 3; <0.001; 0.380
16–24 years old	1316 (31.1) ^a^	95 (13.3) ^b^		94 (16.0)	1 (0.8)	
25–49 years old	1336 (31.6) ^a^	161 (22.5) ^b^		114 (19.4)	47 (36.4)	
50–69 years old	1313 (31.1) ^a^	240 (33.5) ^a^		163 (27.8)	77 (59.7)	
70 years old and more	263 (6.2) ^a^	220 (30.7) ^b^		216 (36.8)	4 (3.1)	
**Educational level**			59.36; 1; <0.001; 0.110			0.32; 1; 0.571; 0.021
No higher education	2088 (49.4)	465 (64.9)		384 (65.4)	81 (62.8)	
Higher education	2140 (50.6)	251 (35.1)		203 (34.6)	48 (37.2)	
**Occupational status**			185.78; 1; <0.001; 0.194			-
Remunerated workforce	2136 (50.5)	165 (23.0)		165 (28.1)	0 (0.0)	
Other	2092 (49.5)	551 (77.0)		422 (71.9)	129 (100.0)	
**Financial situation**			129.12; 1; <0.001; 0.162			71.05; 1; <0.001; 0.315
Perceived as easy	3245 (76.8)	405 (56.6)		375 (63.9)	30 (23.3)	
Perceived as difficult	983 (23.2)	311 (43.4)		212 (36.1)	99 (76.7)	
**Gender**			-			-
Cis Man	2105 (49.8)	303 (42.3)		250 (42.6)	53 (41.1)	
Cis Woman	2098 (49.6)	403 (56.3)		328 (55.9)	75 (58.1)	
Trans Man	3 (0.1)	2 (0.3)		2 (0.3)	0 (0.0)	
Trans Woman	1 (0.0)	0 (0.0)		0 (0.0)	0 (0.0)	
Other	21 (0.5)	8 (1.1)		7 (1.2)	1 (0.8)	
**Sexual orientation**			18.32; 1; <0.001; 0.061			5.01; 1; 0.025; 0.84
SI-heterosexual	3853 (91.1)	616 (86.0)		513 (87.4)	103 (79.8)	
SI-LGB+	375 (8.9)	100 (14.0)		74 (12.6)	26 (20.2)	

^a,b^ The presented proportions per variable with different superscripts differ significantly (post-hoc χ^2^ test *p* > 0.05). Notes: Because the comparisons in this table involved 6 independent tests, we adopted a Bonferroni-corrected significance level of 0.05/6 = 0.008 for these analyses. Abbreviations: SD = Standard Deviation; SI = Self-Identified; LGB+ = lesbian, gay, bisexual, pan-/omnisexual, asexual, other; df = degrees of freedom; V = Cramer’s V.

**Table 2 healthcare-13-03125-t002:** Sample weights. A comparison of the distribution between the Belgian population and the study’s sample.

Age Group	Sex at Birth	Population N	Population Proportion	Sample *n*	Sample Proportion	Population/Sample = Weights
16–24 years old	Female	576,098	0.06	687	0.13	0.46
	Male	601,426	0.06	724	0.15	0.40
25–49 years old	Female	1,864,081	0.20	787	0.16	1.25
	Male	1,883,527	0.20	710	0.14	1.43
50–69 years old	Female	1,475,820	0.16	764	0.15	1.07
	Male	1,458,421	0.15	789	0.16	0.94
70–99 years old	Female	894,533	0.09	279	0.06	1.50
	Male	653,772	0.07	204	0.04	1.75
Total		9,407,678	1.00	4944	1.00	

**Table 3 healthcare-13-03125-t003:** Observed mental health, quality of life, and well-being.

Variable	Within Total Sample (n = 4944)			Within Group Disability (n = 716)		
	No Disability (n = 4228; 85.5%) n (%)	Disability (n = 716; 14.5%) n (%)	χ^2^; df; *p*-Value; V	Able to Work (n = 587; 82.0%) n (%)	Unable to Work Due to Disability (n = 129; 18.0%) n (%)	χ^2^; df; *p*-Value; V
**Quality of life** **[median (IQR)]**	4.0 (4.0–5.0)	4.0 (3.0–5.0)	368.21; 4; <0.001; 0.273			52.68; 4; <0.001; 0.271
Very poor	14 (0.3) ^a^	20 (2.8) ^b^		11 (1.9) ^a^	9 (7.0) ^b^	
Poor	78 (1.8) ^a^	60 (8.4) ^b^		33 (5.6) ^a^	27 (20.9) ^b^	
Neither poor nor good	424 (10.0) ^a^	190 (26.5) ^b^		153 (26.1) ^a^	37 (28.7) ^a^	
Good	2514 (59.5) ^a^	375 (52.4) ^b^		322 (54.9) ^a^	53 (41.1) ^b^	
Very good	1198 (28.3) ^a^	71 (9.9) ^b^		68 (11.6) ^a^	3 (2.3) ^b^	
**Resilience** **[median (IQR)]**	4.0 (2.3–4.0)	3.2 (2.5–4.0)	6.05; 2; 0.049; 0.035			7.73; 2; 0.021; 0.104
Low	1526 (36.1) ^a^	266 (37.2) ^a^		215 (36.6) ^a^	51 (39.5) ^a^	
Normal	2243 (53.1) ^a^	394 (55.0) ^a^		333 (56.7) ^a^	61 (47.3) ^a^	
High	459 (10.9) ^a^	56 (7.8) ^b^		39 (6.6) ^a^	17 (13.2) ^b^	
**Depression** **[median (IQR)]**	3.0 (1.0–6.0)	6.0 (3.0–10.0)	227.44; 4; <0.001; 0.214			38.25; 4; <0.001; 0.231
Minimal	2647 (62.6) ^a^	287 (40.1) ^b^		257 (43.8) ^a^	30 (23.3) ^b^	
Mild	1024 (24.2) ^a^	214 (29.9) ^b^		180 (30.7) ^a^	34 (26.4) ^a^	
Moderate	353 (8.3) ^a^	95 (13.3) ^b^		71 (12.1) ^a^	24 (18.6) ^b^	
Moderately severe	148 (3.5) ^a^	67 (9.4) ^b^		47 (8.0) ^a^	20 (15.5) ^b^	
Severe	56 (1.3) ^a^	53 (7.4) ^b^		32 (5.5) ^a^	21 (16.3) ^b^	
**Anxiety** **[median (IQR)]**	4.0 (1.0–7.0)	5.0 (2.0–10.0)	103.62; 3; <0.001; 0.145			19.62; 3; <0.001; 0.166
Minimal	2463 (58.3) ^a^	326 (45.5) ^b^		289 (49.2) ^a^	37 (28.7) ^b^	
Mild	1233 (29.2) ^a^	209 (29.2) ^a^		163 (27.8) ^a^	46 (35.7) ^a^	
Moderate	346 (8.2) ^a^	92 (12.8) ^b^		71 (12.1) ^a^	21 (16.3) ^a^	
Severe	186 (4.4) ^a^	89 (12.4) ^b^		64 (10.9) ^a^	25 (19.4) ^b^	
**PTSD** **[median (IQR)]**	0.0 (0.0–0.0)	0.0 (0.0–0.1)	50.38; 1; <0.001; 0.101			22.52; 1; <0.001; 0.177
No PTSD	3838 (90.8)	587 (82.0)		500 (85.2)	87 (67.4)	
Probable PTSD	390 (9.2)	129 (18.0)		87 (14.8)	42 (32.6)	
**Hazardous alcohol use**			52.37; 1; <0.001; 0.103			0.00; 1; 0.948; 0.002
No	2593 (61.3)	540 (75.4)		443 (75.5)	97 (75.2)	
Yes	1635 (38.7)	176 (24.6)		144 (24.5)	32 (24.8)	
**Sedative use**			277.62; 2; <0.001; 0.237			11.58; 2; 0.003; 0.127
No	2939 (69.5) ^a^	301 (42.0) ^b^		264 (45.0) ^a^	37 (28.7) ^b^	
Lifetime	569 (13.5) ^a^	103 (14.4) ^a^		81 (13.8) ^a^	22 (17.1) ^a^	
Past 12 months	720 (17.0) ^a^	312 (43.6) ^b^		242 (41.2) ^a^	70 (54.3) ^b^	
**Cannabis use**			31.94; 2; <0.001; 0.080			2.14; 2; 0.342; 0.055
No	3194 (75.5) ^a^	607 (84.8) ^b^		503 (85.7) ^a^	104 (80.6) ^a^	
Lifetime	611 (14.5) ^a^	54 (7.5) ^b^		42 (7.2) ^a^	12 (9.3) ^a^	
Past 12 months	423 (10.0) ^a^	55 (7.7) ^a^		42 (7.2) ^a^	13 (10.1) ^a^	
**Illegal drug use**			1.51; 2; 0.471; 0.017			3.69; 2; 0.158; 0.072
No	3964 (93.8) ^a^	678 (94.7) ^a^		560 (95.4) ^a^	118 (91.5) ^a^	
Lifetime	157 (3.7) ^a^	20 (2.8) ^a^		15 (2.6) ^a^	5 (3.9) ^a^	
Past 12 months	107 (2.5) ^a^	18 (2.5) ^a^		12 (2.0) ^a^	6 (4.7) ^a^	
**Suicide attempt**			75.46; 2; <0.001; 0.124			6.54; 2; 0.038; 0.096
No	4018 (95.0) ^a^	620 (86.6) ^b^		516 (87.9) ^a^	104 (80.6) ^b^	
Lifetime	179 (4.2) ^a^	80 (11.2) ^b^		61 (10.4) ^a^	19 (14.7) ^a^	
Past 12 months	31 (0.7) ^a^	16 (2.2) ^b^		10 (1.7) ^a^	6 (4.7) ^b^	
**Self-harm**			7.13; 2; 0.028; 0.038			4.02; 2; 0.134; 0.075
No	3806 (90.0)	623 (87.0)		514 (87.6) ^a^	109 (84.5) ^a^	
Lifetime	299 (7.1)	61 (8.5)		51 (8.7) ^a^	10 (7.8) ^a^	
Past 12 months	123 (2.9)	32 (4.5)		22 (3.7)^a^	10 (7.8) ^b^	

^a,b^ The presented proportions per variable with different superscripts differ significantly (post-hoc χ^2^ test *p* > 0.05). Note: A corrected *p*-level of 0.05/11 = 0.004 was used as the critical significance level for both sets of comparisons. Abbreviations: PTSD = Post Traumatic Stress Disorder; IQR = InterQuartile Range; df = degrees of freedom; V = Cramer’s V.

**Table 4 healthcare-13-03125-t004:** Lifetime sexual victimization.

Variable	Within Total Sample (n = 4944)			Within Group Disability (n = 716)		
	No Disability (n = 4228; 85.5%) n (%)	Disability (n = 716; 14.5%) n (%)	χ^2^; df; *p*-Value; V	Able to Work (n = 587; 82.0%) n (%)	Unable to Work Due to Disability (n = 129; 18.0%) n (%)	χ^2^; df; *p*-Value; V
**Any SV**	2635 (62.3)	432 (60.3)	1.027; 1; 0.311; 0.014	346 (58.9)	86 (66.7)	2.64; 1; 0.104; 0.061
**Any Hands-Off SV**	2418 (57.2)	382 (53.4)	3.67; 1; 0.055; 0.027	302 (51.4)	80 (62.0)	4.74; 1; 0.029; 0.081
Sexual staring	1610 (38.1)	240 (33.5)	5.44; 1; 0.020; 0.033	183 (31.2)	57 (44.2)	8.03; 1; 0.005; 0.106
Sexual innuendo	1421 (33.6)	214 (29.9)	3.85; 1; 0.050; 0.028	162 (27.6)	52 (40.3)	8.16; 1; 0.004; 0.107
Showing sexual images	719 (17.0)	122 (17.1)	0.00; 1; 0.970; 0.001	93 (15.9)	29 (22.5)	3.26; 1; 0.071; 0.068
Sexual calls or texts	503 (11.9)	86 (12.0)	0.01; 1; 0.922; 0.001	67 (11.4)	19 (14.7)	1.08; 1; 0.298; 0.039
Voyeurism	106 (2.5)	21 (2.9)	0.46; 1; 0.498; 0.010	12 (2.1)	9 (7.0)	8.98; 1; 0.003; 0.112
Distributing sexual images	62 (1.5)	13 (1.8)	0.50; 1; 0.481; 0.010	10 (1.7)	3 (2.3)	0.713 °
Exhibitionism	575 (13.6)	115 (16.1)	3.08; 1; 0.079; 0.025	89 (15.2)	26 (20.2)	1.96; 1; 0.162; 0.052
Forcing to show intimate body parts	222 (5.3)	46 (6.4)	1.66; 1; 0.197; 0.018	31 (5.3)	15 (11.6)	7.05; 1; 0.008; 0.099
**Any Hands-On SV**	1241 (29.4)	271 (37.8)	20.82; 1; <0.001; 0.065	214 (36.5)	57 (44.2)	2.69; 1; 0.101; 0.061
**Any Sexual Abuse**	1142 (27.0)	248 (34.6)	17.62; 1; <0.001; 0.060	195 (33.2)	53 (41.1)	2.89; 1; 0.089; 0.064
Kissing	658 (15.6)	141 (19.7)	7.71; 1; 0.005; 0.039	113 (19.3)	28 (21.7)	0.40; 1; 0.526; 0.024
Touching in care	274 (6.5)	75 (10.5)	14.89; 1; <0.001; 0.055	55 (9.4)	20 (15.5)	4.24; 1; 0.039; 0.077
Fondling/rubbing	621 (14.7)	144 (20.1)	13.77; 1; <0.001; 0.053	109 (18.6)	35 (27.1)	4.83; 1; 0.028; 0.082
Forced undressing	158 (3.7)	51 (7.1)	17.34; 1; <0.001; 0.059	34 (5.8)	17 (13.2)	8.72; 1; 0.003; 0.110
**Any Rape**	398 (9.4)	111 (15.5)	24.58; 1; <0.001; 0.071	81 (13.8)	30 (23.3)	7.22; 1; 0.007; 0.100
Oral penetration	140 (3.3)	46 (6.4)	16.39; 1; <0.001; 0.058	31 (5.3)	15 (11.6)	7.09; 1; 0.008; 0.099
Attempt at oral penetration	151 (3.6)	36 (5.0)	3.57; 1; 0.059; 0.027	24 (4.1)	12 (9.3)	6.02; 1; 0.014; 0.092
Vaginal or anal penetration	172 (4.1)	57 (8.0)	21.00; 1; <0.001; 0.065	41 (7.0)	16 (12.4)	4.24; 1; 0.040; 0.077
Attempt of vaginal or anal penetration	116 (2.7)	33 (4.6)	7.33; 1; 0.007; 0.039	20 (3.4)	13 (10.1)	10.67; 1; 0.001; 0.122
Forcing to penetrate	35 (0.8)	14 (2.0)	7.93; 1; 0.005; 0.040	10 (1.7)	4 (3.1)	0.294 °

° Fisher’s exact test. Notes: Because the comparisons in this table involved 6 independent tests, we adopted a Bonferroni-corrected significance level of 0.05/22 = 0.002 for these analyses. Abbreviations: SV = Sexual Violence; df = degrees of freedom; V = Cramer’s V.

**Table 5 healthcare-13-03125-t005:** Logistic Regression Analysis of the Total Sample for Two Outcome Variables: Prevalence of Hands-off Sexual Violence and Hands-on Sexual Violence.

	Hands-Off Sexual Violence			Hands-On Sexual Violence		
Predictors	EXP (B) Odds Ratio	95% C.I. Odds Ratio (Wald)	*p*-Value (LRT)	EXP (B) Odds Ratio	95% C.I. Odds Ratio (Wald)	*p*-Value (LRT)
**Sex assigned at birth** (ref. Male)			<0.001			<0.001
Female	4.77	4.16–5.47		2.88	2.50–3.31	
**Age** (ref. 16–24 years old)			<0.001			<0.001
25–49 years old	0.83	0.69–1.00		0.81	0.67–0.99	
50–69 years old	0.57	0.47–0.69		0.99	0.81–1.21	
70 years old and more	0.40	0.31–0.53		1.12	0.85–1.48	
**Educational level** (ref. No higher education)			0.461			0.100
Higher education	1.08	0.94–1.25		1.23	1.06–1.42	
**Financial situation** (ref. Perceived as easy)			<0.001			<0.001
Perceived as difficult	1.08	0.92–1.27		1.08	0.92–1.26	
**Sexual orientation** (ref. SI-Heterosexual)			<0.001			<0.001
SI-LGB+	1.50	1.18–1.92		1.38	1.11–1.72	
**Disability** (ref. No)			0.912			<0.001
Disability, but not incapacitated to work	0.82	0.65–1.02		1.24	0.99–1.54	
Disability and incapacitated to work	0.77	0.49–1.20		1.24	0.82–1.88	
**Quality of Life**	1.14	1.03–1.27	0.061	1.06	0.96–1.18	0.009
**Resilience**	1.06	1.00–1.13	0.367	1.05	0.99–1.12	0.201
**Depression**	1.02	1.00–1.04	<0.001	1.01	0.99–1.03	<0.001
**Anxiety**	1.05	1.03–1.07	<0.001	1.03	1.01–1.05	<0.001
**PTSD**	1.28	1.20–1.37	<0.001	1.23	1.16–1.30	<0.001
**Hazardous alcohol use** (ref. no)			<0.001			<0.001
Yes	1.23	1.07–1.42		1.35	1.17–1.55	
**Sedative use** (ref. no)			<0.001			<0.001
Lifetime, but not past 12 months	1.57	1.28–1.93		1.28	1.05–1.55	
Past 12 months	1.04	0.87–1.25		1.11	0.93–1.32	
**Cannabis use** (ref. no)			<0.001			<0.001
Lifetime, but not past 12 months	1.79	1.45–2.21		2.08	1.70–2.54	
Past 12 months	1.85	1.42–2.42		1.75	1.36–2.24	
**Illegal drug use** (ref. no)			0.216			0.104
Lifetime, but not past 12 months	1.08	0.74–1.58		1.19	0.83–1.69	
Past 12 months	1.45	0.90–2.40		1.43	0/94–2.19	
**Suicide attempt** (ref. no)			<0.001			<0.001
Lifetime, but not past 12 months	1.54	1.07–2.24		1.59	1.18–2.15	
Past 12 months	2.11	0.95–4.92		1.76	0.88–3.51	
**Self-harm** (ref. no)			<0.001			<0.001
Lifetime, but not past 12 months	2.02	1.45–2.84		1.67	1.29–2.16	
Past 12 months	0.91	0.57–1.48		1.09	0.73–1.63	

Abbreviations: LRT = Likelihood Ratio Test; ref = reference category; SI = Self-Identified; LGB+ = Lesbian, Gay, Bisexual, pan-/omnisexual, asexual, other; PTSD = Post Traumatic Stress Disorder.

## Data Availability

The data supporting the findings of this study contain confidential and sensitive information and therefore cannot be shared publicly. Access to the data may be granted upon reasonable request to the corresponding author, contingent upon compliance with applicable data protection regulations and approval by the relevant ethics committee.

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
