# Peer review of "Addressing Sexual Violence Against Persons with Disabilities in Belgium"

_healthcare, 2025, doi:10.3390/healthcare13233125_

Round 1

Reviewer 1 Report

Comments and Suggestions for Authors

The authors refer to structural vulnerabilities in the Introduction, yet they do not provide a clear definition or examples of what this concept entails. It is recommended that the authors clarify this term, for instance by offering examples in parentheses and referencing relevant literature in which structural vulnerabilities have been identified. As this concept plays a critical role in the work, such clarification is essential.

In line 125, the reference to a mixed type of disability may cause confusion, as it is unclear what the authors intend by this term. This ambiguity becomes particularly problematic when disability is discussed alongside chronic illness. I suggest that the authors consult the World Health Organization’s definition of disability or the World Report on Disability, which does not conflate disability and chronic illness. The authors should explicitly identify the types or categories of disability being examined, as well as clearly define what is meant by chronic illness. Although individuals with disabilities may also experience chronic illness, these are conceptually distinct constructs. The authors should therefore outline disability and chronic illness as separate concepts and only then specify the theoretical perspective from which the manuscript addresses their relationship.

I recommend that section 2.2.3 be rewritten to reflect these clarifications.

In the Discussion section, lines 415–419 should also be revised. The authors should refer more precisely to specific factors and to the process of othering. Additionally, the manuscript should avoid conflating chronic

Author Response

Comment 1: The authors refer to structural vulnerabilities in the Introduction, yet they do not provide a clear definition or examples of what this concept entails. It is recommended that the authors clarify this term, for instance by offering examples in parentheses and referencing relevant literature in which structural vulnerabilities have been identified. As this concept plays a critical role in the work, such clarification is essential.

Response 1: We thank the reviewer for this valuable observation. The examples illustrating structural vulnerabilities were already included in the text. However, we agree that these conditions should have been explicitly identified and defined as structural vulnerabilities earlier in the Introduction. We have now added a concise definition and clarification to make this conceptual link explicit (lines 118–123).

Comment 2: In line 125, the reference to a mixed type of disability may cause confusion, as it is unclear what the authors intend by this term. This ambiguity becomes particularly problematic when disability is discussed alongside chronic illness. I suggest that the authors consult the World Health Organization’s definition of disability or the World Report on Disability, which does not conflate disability and chronic illness. The authors should explicitly identify the types or categories of disability being examined, as well as clearly define what is meant by chronic illness. Although individuals with disabilities may also experience chronic illness, these are conceptually distinct constructs. The authors should therefore outline disability and chronic illness as separate concepts and only then specify the theoretical perspective from which the manuscript addresses their relationship.

I recommend that section 2.2.3 be rewritten to reflect these clarifications.

Response 2: We thank the reviewer for this valuable observation. We fully agree that disability and chronic illness are conceptually distinct. However in our study design and survey instrument both concepts were assessed through self-reported items without providing participants with strict definitions. In practice, many respondents appeared to use the terms interchangeably, and many open-text answers described conditions that could not be reliably classified as either “disability” or “chronic illness”. Given this overlap, and because the broader UN-MENAMAIS study was not originally designed to examine persons with disabilities as a primary focus, we opted to merge the two variables into one inclusive category (“disability and/or chronic illness”) representing individuals who reported functional limitations in daily life. This pragmatic approach captures a wider spectrum of participants who may experience structural barriers and vulnerabilities relevant to the study’s objectives. We have now clarified this methodological choice in Section 2.2.3. We have also added an explicit explanation of this issue, and its implications for interpretation, to the Limitations section.

Additionally, to avoid any ambiguity, we have specified in the Introduction (lines 130–133) what is meant by a “mixed type” of disability, namely individuals who experience more than one form of impairment (e.g., a combination of physical and cognitive limitations).

Comment 3: In the Discussion section, lines 415–419 should also be revised. The authors should refer more precisely to specific factors and to the process of othering. Additionally, the manuscript should avoid conflating chronic

Response 3: We thank the reviewer for this helpful suggestion. We have revised the text accordingly in the second paragraph of the Discussion, to explicitly link the process of othering and ableism to specific mechanisms reflected in our findings. The paragraph now highlights that poorer mental health, financial hardship, social isolation, and dependency on others for care or housing can increase vulnerability to sexual violence. These interrelated disadvantages are discussed as outcomes of structural and social processes that marginalize PwD and limit access to economic stability, autonomy, and support.

Reviewer 2 Report

Comments and Suggestions for Authors

COMMENT 1 (lines 154-176): Differences in data collection strategies for different age ranges – e.g., online surveys (16- to 69 years) and structured face-to face interviews (70+ years) – have the potential to influence prevalence rates (and values for associated variables). This issue needs to be addressed more directly in the text and/or in the study limitations section.

COMMENT 2 (lines 228-230): The ‘subjective’ assessment of PwD could be viewed as a possible study limitation, with this needing to be more fully acknowledged in the study limitations section.

COMMENT 3 (lines 231-236). PwD may not have full and equal participation in society due to a number of associated functional impairments. Please provide a clearer rationale for why ‘able to work’ was selected as a primary index of functional impairment in this study.

COMMENT 4 (lines 274-286): It is possible (if not likely) that individuals with intellectual (and possibly mental) forms of PwD will vary in their capacity to provide truly informed consent. Was this issue addressed in the study – if so (please clarify how), if not please consider this as a potential limitation of the study.

COMMENT 5: (Discussion section): It would be informative in the discussion section for the authors to compare obtained prevalence rates with prevalence rates reported in previous studies conducted in Western Europe (and, indeed, in other parts of the world) – see, for example, Mailhot Amborski et al., 2022 (cited by the authors).

Author Response

COMMENT 1 (lines 154-176): Differences in data collection strategies for different age ranges – e.g., online surveys (16- to 69 years) and structured face-to face interviews (70+ years) – have the potential to influence prevalence rates (and values for associated variables). This issue needs to be addressed more directly in the text and/or in the study limitations section.

Response 1: We thank the reviewer for this valuable comment. We have added a sentence to Section 4.1 (Limitations and Suggestions for Future Research) acknowledging that differences in data collection strategies between age groups (online vs. face-to-face) may have influenced reporting patterns and comparability of prevalence rates.

COMMENT 2 (lines 228-230): The ‘subjective’ assessment of PwD could be viewed as a possible study limitation, with this needing to be more fully acknowledged in the study limitations section.

Response 2: We agree with the reviewer’s observation. We have expanded Section 4.1 to explicitly recognize that disability status was based on self-assessment rather than clinical or administrative verification, which could introduce reporting bias.

COMMENT 3 (lines 231-236). PwD may not have full and equal participation in society due to a number of associated functional impairments. Please provide a clearer rationale for why ‘able to work’ was selected as a primary index of functional impairment in this study.

Response 3: We have clarified our rationale for selecting “able to work” as an indicator of functional impairment in Section 2.2.3. In the Belgian context, being unable to work (work incapacity) is often linked to medical evaluation and may reflect more objectively verifiable limitations in functioning. Although our survey data do not allow us to confirm whether each respondent’s inability to work was formally certified by a physician, this variable offers a pragmatic proxy for functional impairment that goes beyond self-perceived disability. Considering that our group of persons with disabilities (PwD) is highly heterogeneous and based on self-reported conditions, this operationalization allowed us to distinguish a subgroup likely to experience more pronounced, externally validated limitations from those whose disability may be more subjectively experienced. We have revised the text accordingly to reflect this nuance.

COMMENT 4 (lines 274-286): It is possible (if not likely) that individuals with intellectual (and possibly mental) forms of PwD will vary in their capacity to provide truly informed consent. Was this issue addressed in the study – if so (please clarify how), if not please consider this as a potential limitation of the study.

Response 4: We appreciate this important ethical consideration. We have revised Section 4.1 to clarify that participants’ capacity to provide informed consent was not explicitly assessed in the online survey among respondents aged 16–69 years, which may represent a potential ethical and methodological limitation. However, for the structured face-to-face interviews with participants aged 70 years and older, interviewers were trained to assess understanding and consent capacity prior to participation. While we consider the overall likelihood of compromised consent to be low given these procedures, we have acknowledged this as a limitation in the revised manuscript.

COMMENT 5: (Discussion section): It would be informative in the discussion section for the authors to compare obtained prevalence rates with prevalence rates reported in previous studies conducted in Western Europe (and, indeed, in other parts of the world) – see, for example, Mailhot Amborski et al., 2022 (cited by the authors).

Response 5: We thank the reviewer for this helpful suggestion. In response, we have expanded the beginning of the Discussion section to situate our findings within the broader international literature. Specifically, we now clarify that prevalence estimates of SV among PwD vary widely across studies due to differences in disability types examined and definitions of SV used, making direct comparison difficult. We additionally highlight that, although our results confirm the overall trend of increased vulnerability among PwD, the magnitude of the difference in our sample is smaller than the approximately twofold elevated risk documented in the meta-analysis by Mailhot Amborski et al. (2022).

Reviewer 3 Report

Comments and Suggestions for Authors

The manuscript needs the following clarifications:

  1. Table 3: Anxiety, Depression & PTSD: Mean and SD values are quite close to each other, which signifies that the dataset was either skewed or contained outliers. In this situation, it is better to represent as median and IQR, instead of mean and SD.
  2. Please explain a little bit about the logistic regression output as running text – whether the model was fit or not? How much variation of the dependent variable could be explained by the independent variables?
  3. Moreover, how the independent variables were chosen in the regression model needs to be explained.
  4. Line 168: “a cluster random probability sampling” – could you please explain a little bit about the different clusters? What comprised the clusters?
  5. Line 299: “To avoid multicollinearity, the correlations were checked between” – which type of corelation was assessed?
  6. Table 1: What was the operational definition of “No higher education” – clarify for better understanding of the international readers.
  7. Table 3: What was the operational definition of low, normal and high resilience? – Please cite reference as well.
  8. Similarly, mention the definition of different grades of quality of life.

Author Response

Comment 1: Table 3: Anxiety, Depression & PTSD: Mean and SD values are quite close to each other, which signifies that the dataset was either skewed or contained outliers. In this situation, it is better to represent as median and IQR, instead of mean and SD.

Response 1: We thank the reviewer for this accurate observation. We fully agree that mean and SD are not the most appropriate descriptive statistics when the data are skewed or contain outliers. As suggested, we have replaced the mean and SD values with the median and interquartile range (IQR) in Table 3. The table has been updated accordingly.

Comment 2: Please explain a little bit about the logistic regression output as running text – whether the model was fit or not? How much variation of the dependent variable could be explained by the independent variables?

Response 2: We thank the reviewer for this comment. As now clarified in Section 2.5 (Analysis), the logistic regression models were not used for predictive purposes or formal model fitting but rather to explore associations and identify patterns consistent with known vulnerability factors. For this reason, model optimisation procedures (e.g., stepwise selection based on fit indices) were not applied. Nonetheless, we have now added the explained variance (R²) for both models in the Results section: the predictors accounted for 22.8% of the variance in hands-off SV and 14.2% in hands-on SV. These additions should provide the reviewer with a clearer understanding of the models' explanatory power.

Comment 3: Moreover, how the independent variables were chosen in the regression model needs to be explained.

Response 3: The selection of independent variables was theory-driven and grounded in existing literature identifying key vulnerability factors for sexual victimization, including socio-demographic characteristics, socio-economic position, mental health indicators, and coping behaviours. These variables align with well-established correlates of sexual violence risk and with the framework of structural vulnerabilities discussed in the Introduction. Additionally, variable selection was guided by our research questions, which aimed to examine whether the general risk factors for sexual victimization also apply to persons with disabilities and whether disability moderates these associations. This information was added in the 2.4. Analysis section.

Comment 4: Line 168: “a cluster random probability sampling” – could you please explain a little bit about the different clusters? What comprised the clusters?

Response 4: We have now clarified what the clusters comprised by adding a brief explanation on lines 177–178. For readers interested in a full methodological description of the sampling design and the random-walk recruitment procedure, we warmly refer to the original Belgian prevalence study on sexual violence in older adults: Nobels, A., Cismaru Inescu, A., Nisen, L., Hahaut, B., Beaulieu, M., Lemmens, G., Adam, S., Schapansky, E., Vandeviver, C., & Keygnaert, I. (2021). Sexual violence in older adults: A Belgian prevalence study. https://doi.org/10.1186/s12877-021-02623-x

Comment 5: Line 299: “To avoid multicollinearity, the correlations were checked between” – which type of corelation was assessed?

Response 5: Multicollinearity was assessed using Variance Inflation Factors (VIFs) rather than simple pairwise correlations. All VIF values were well below commonly accepted cut-offs, indicating no issues with multicollinearity.

Comment 6: Table 1: What was the operational definition of “No higher education” – clarify for better understanding of the international readers.

Response 6: The operational definition of “No higher education” has now been clarified in the manuscript. Specifically, we added an explicit explanation on line 203, stating that higher education refers to college or university programmes leading to a bachelor’s or master’s degree.

Comment 7: Table 3: What was the operational definition of low, normal and high resilience? – Please cite reference as well.

Similarly, mention the definition of different grades of quality of life.

Response 7: The Brief Resilience Scale (with reference) as well as the definitions of the different quality-of-life score ranges are already described in detail in Section 2.2.4 of the Methods. To avoid overloading Table 3 with extensive scale information, we chose to keep these descriptions in the Methods section. Including these definitions directly in the table would require doing the same for all other mental health variables (e.g., depression, anxiety, PTSD), which would considerably reduce the table’s readability.

Round 2

Reviewer 2 Report

Comments and Suggestions for Authors

No comments

Reviewer 3 Report

Comments and Suggestions for Authors

The authors have successfully revised all the previous queries. The manuscript can now be accepted for final publication. Good work done by the authors.